# The Impact of Noncontact Tonometry and Icare Rebound Tonometry on Tear Stability and Dry Eye Clinical Practice

**DOI:** 10.3390/jcm11102819

**Published:** 2022-05-17

**Authors:** Murat Dogru, Cem Simsek, Takashi Kojima, Naohiko Aketa, Kazuo Tsubota, Jun Shimazaki

**Affiliations:** 1Department of Ophthalmology, Keio University School of Medicine, Tokyo 160-8582, Japan; cemsimsekmd@gmail.com (C.S.); kojkoj@me.com (T.K.); nao.nao.pao.pao@gmail.com (N.A.); tsubota@tsubota-lab.com (K.T.); 2Department of Ophthalmology, Mugla Sitki Kocman University School of Medicine, Mugla 48000, Turkey; 3Tsubota Laboratory, Inc., Tokyo 160-0016, Japan; 4Department of Ophthalmology, Tokyo Dental College, Chiba 261-8502, Japan; meishano1@gmail.com

**Keywords:** dry eye, IOP measurement, tear stability, tonometer

## Abstract

The purpose of this study was to investigate the possible effects of the noncontact air puff tonometry (NCT) and Icare rebound tonometry (ICT) on the tear film stability by using the tear stability analysis system (TSAS) and dry eye parameters. Fifteen eyes from fifteen normal healthy subjects were investigated in this study. All subjects underwent TSAS surface regularity index (SRI) examinations, TBUT, and IOP measurements. The mean IOP results measured with NCT were 13.3 ± 1.86 mm Hg, and the mean IOP results measured with ICT were 15.88 ± 3.09 mm Hg (*p* > 0.05). The mean values of baseline, 5 min, and 10 min of the NCT-SRI and ICR-SRI were tested. There were statistically significant differences between NCT-Baseline SRI, NCT-5 min SRI, and NCT-10 min SRI values (*p* < 0.05). SRI values significantly increased after NCT. The mean values of the baseline, 5 min, and 10 min of the ICT-SRI were also assessed. There were no statistically significant differences between ICT-Baseline SRI, ICT-5 min SRI, and ICT-10 min SRI values (*p* > 0.05). The mean TBUT values exhibited a significant decrease at 1 min, 5 min, and 10 min compared with baseline values for the NCT and ICT (*p* < 0.01). NCT-TBUT and ICT-TBUT values were also compared with each other in the same time period. There were no statistically significant differences between NCT-Baseline and ICT-Baseline TBUT values (*p* > 0.05). In conclusion, intraocular pressure measurements in routine ophthalmology clinical practices by either NCT or ICT cause deterioration in the tear film stability which might affect tear stability testing when performed soon after IOP measurements. It is best to wait at least for 20–30 min after the IOP measurement before evaluating the tear film and the corneal surface or perform tonometry after the tear film-ocular surface evaluation tests.

## 1. Introduction

Intraocular pressure (IOP) measurement is one of the most essential examination parameters used in routine ophthalmic practice. Accurate and consistent measurement of IOP is essential not only for glaucoma, but also for the diagnosis and postsurgical follow-up of corneal, lenticular and vitreoretinal diseases [1]. Many methods, especially those involving tonometers, have been developed to evaluate the IOP. The Goldmann Applanation Tonometer (GAT), developed in 1955, has been accepted as the gold standard method for examining patients with glaucoma in clinical practice [2]. However, the GAT necessitates a slit lamp microscope, and the use of topical anesthetics with fluorescein dye. Therefore, practical measurement methods have been developed, such as air puff tonometry using noncontact tonometry (NCT) [3]. On the other hand, Icare rebound tonometry (ICT) is a novel technique that has been in clinical practice since 1996, does not require topical anesthesia, and is used especially for intraocular pressure measurement in bedridden and pediatric patients [4,5]. While measuring IOP with the ICT, there is minimal contact with the cornea for a very short time.

The 2017 TFOS Dry Eye Work Shop II report emphasized the importance of an unstable tear film in the diagnosis of dry eyes and the non-invasive measurement of tear film break-up time (TBUT) (when possible) as an essential diagnostic test. The TBUT is useful for assessing the tear film stability and is widely used in almost all ophthalmology clinics. However, diagnostic methods delivering fluorescein dye into the tear film induce reflex tearing and are regarded as invasive [6,7,8,9]. Therefore, technologies such as grid keratometry, DR-1 tear film lipid layer interferometry or the tear stability analysis system (TSAS), which can evaluate the tear film stability non-invasively, are regarded as helpful tear stability assessment methods. The TSAS can analyze the surface regularity index (SRI) and surface asymmetry index (SAI), which have been shown to be significantly worse in dry eyes as compared with health subjects [7].

To the best of our knowledge, the possible effects of noncontact air puff tonometry and the ICT on the tear film stability have not been investigated so far. In this study, we investigated the alterations to tear film stability after air puff and Icare rebound tonometry by using TSAS and TBUT and the possible impact of these tonometry tests on dry eye specialty clinical practices. 

## 2. Materials and Methods

This prospective study was approved by the institutional review board of Tokyo Dental College. The study adhered to the tenets of the Declaration of Helsinki, and consent forms were obtained from all subjects after an explanation of the purpose and process of the study was provided.

### 2.1. Subjects and Procedures

Fifteen eyes from fifteen normal healthy subjects (eight males, seven females; mean age: 35.4, age range: 32–40 years) recruited from Tokyo Dental College, Department of Ophthalmology (Ichikawa, Japan) were investigated in this study. Standard noncontact tonometry (Topcon TRK-2P, Tokyo, Japan) and Icare 100 tonometry (Tiolat Oy, Helsinki, Finland) were performed for the participants. All subjects underwent tear stability analysis system (TSAS) (SRI) examinations, TBUT and IOP measurements. Subjects included in this study were asymptomatic, with no history of contact lens use, no history of eye surgery or topical or systemic drug use. Subjects underwent OSDI questionnaires, TBUT and fluorescein and Lissamine Green vital stainings for recruitment into the study. Only asymptomatic subjects with OSDI scores < 13 points, with vital staining scores < 1 point and TBUT scores > 5 s were recruited.

On the day of the tonometry measurements, the tonometry device and the TSAS device were placed next to the slit lamp to ensure rapid patient examinations. The noninvasive TSAS examination was performed initially. The TSAS examination was followed by standard TBUT testing using fluorescein dye. Two hours after TBUT testing, noncontact tonometry was performed. 

The TSAS examination was again performed at 5 min and 10 min after noncontact tonometry. Following the TSAS, TBUT was reassessed at one, five and ten minutes. On the same day, six hours later, the same examinations were performed in the same order using Icare rebound tonometry. The tests were performed in the same room with the humidity set at 42 ± 3% and temperature at 24 ± 3 °C. The recruited subjects were required to sit next to each other just outside the examination room. All measurements were completed in a single day. The subjects visited the outpatient clinic in the morning and were required to wait until the afternoon examinations at a room on the same floor where the humidity and temperature settings were the same as the outpatient clinic.

### 2.2. IOP Measurements

#### 2.2.1. Air-Puff Tonometry

A standard noncontact tonometer (Topcon TRK-2P, Tokyo, Japan) was used to measure the IOP.

#### 2.2.2. Icare ic100: Rebound Tonometer

The Icare ic100 tonometer is a portable rebound tonometer that relies on the induction or impact principle. Mainly, the device consists of six parts, including a forehead support, forehead adjusting wheel, probe, display, navigation button and measure button. The Icare tonometry stainless-steel probe is 50 mm long and includes 2 coaxial magnet systems with a diameter of 1.4–1 mm. In order to obtain an accurate measurement, the tonometer was positioned at 4 mm horizontally from the central cornea. The voltage formed in the magnetic system by the corneal contact with the movement of the probe was detected by the sensor and transformed into a digital signal. The probe tip is covered with a plastic to minimize the risk of corneal damage. Six consecutive measurements were taken, and the mean of these six measurements was automatically calculated. 

### 2.3. Tear Stability Assessment

#### 2.3.1. Tear Stability Analysis System Measurements

We evaluated the tear film stability using the tear stability analysis system (TSAS). TSAS measurements were performed for the dynamic assessment of tear stability by topographic analysis (TMS-2N; Tomey Technology, Nagoya, Japan), as reported previously [6,7]. The tear stability regularity index (SRI) was recorded during a testing period of ten seconds [7].

#### 2.3.2. Tear Film Break-Up Time (TBUT)

Initially, 2 μL of 1% fluorescein solution was instilled into the tear meniscus using a micropipette without touching the ocular surface and eyelids. The subjects were asked to blink several times. The tear film was examined at the slit lamp with a broad beam using a cobalt blue filter. The interval between the last blink and the appearance of the first randomly distributed black spot was measured by a stopwatch. This procedure was performed three times consecutively, and the mean of these results was recorded.

## 3. Results

### 3.1. Baseline Tear Functions

The mean OSDI score at the time of recruitment into the study was 2.26 ± 2.63 points (minimum: 0 point; maximum: 8 points). The mean baseline TBUT at the time of recruitment was 6.92 ± 0.50 s for the right eyes and 6.03 ± 0.50 s for the left eyes, respectively. There were no statistically significant differences between the baseline TBUT values between both eyes (*p* > 0.05). All eyes had no fluorescein or Lissamine green staining (0 point score) at the baseline measurements, which allowed them to be included into the study.

### 3.2. IOP Measurements

All IOP measurements with the ICT and NCT were performed by the same experienced clinician. The mean IOP results measured with NCT were 13.3 ± 1.86 mm Hg (range: 10 to 15 mm Hg). The mean IOP results measured with ICT were 15.88 ± 3.09 mm Hg (range 9 to 19 mm Hg). There were no significant differences in terms of mean IOP results between the two tonometry examinations for subjects in the current study (*p* > 0.05). 

### 3.3. Comparison of TSAS Parameters between the NCT and ICT Examinations

The mean SRI results were obtained by TSAS examinations, and are presented in Table 1. The mean values of baseline, 5 min and 10 min of the NCT-SRI were tested with one-way ANOVA tests. There were statistically significant differences between NCT-Baseline SRI, NCT-5 min SRI and NCT-10 min SRI values (*p* < 0.05). SRI values increased significantly after NCT. 

The mean values of the baseline, 5 min and 10 min of the ICT-SRI were also assessed with one-way ANOVA tests. There was no statistically significant difference between ICT-Baseline SRI, ICT-5 min SRI and ICT-10 min SRI values (*p* > 0.05) (Table 1). 

### 3.4. Comparison of TBUT Results between the NCT and ICT Examinations

The mean TBUTs for the NCT were 7.04 ± 1.51 s (second) before IOP measurement (baseline), 2.34 ± 1.41 s after 1 min of IOP measurement, 2.35 ± 1.39 s after 5 min of IOP measurement, and 2.33 ± 1.32 s after 10 min of IOP measurement. The mean TBUT values showed a significant decrease at 1 min, 5 min and 10 min compared with baseline values for the NCT (*p* < 0.01). The mean TBUTs for the ICT were 7.19 ± 2.15 s before IOP measurement, 3.55 ± 1.53 s after 1 min of IOP measurement, 3.41 ± 1.48 s after 5 min of IOP measurement, and 3.74 ± 2.10 s after 10 min of IOP measurement. The mean TBUTs showed a significant decrease at 1 min, 5 min and 10 min compared with baseline values for the ICT (*p* < 0.01). NCT-TBUT and ICT-TBUT values were also comparable with each other in the same time period. There were no statistically significant differences between NCT-Baseline and ICT-Baseline TBUT values (*p* > 0.05). However, statistically significant differences were found for 1 min, 5 min and 10 min TBUT values between NCT and ICT (*p* < 0.05) (Figure 1).

The mean change in TBUT from baseline to 10 min after tonometry (ΔTBUT) was significantly less for ICT measurements compared with NCT measurements (*p* < 0.05), as shown in Figure 2: 53% of the eyes attained a TBUT value of >5 s 10 min after ICT measurements, whereas only 20% of the eyes had a TBUT measurement exceeding 5 s 10 min after NCT measurement.

## 4. Discussion

The noncontact “air puff” tonometer is widely and almost routinely used in ophthalmic clinical practices for measuring IOP. These noncontact tonometers work with the ‘applanation method’ and use air to flatten the cornea in a uniform area. The air puff tonometer sends a pulse of air by blasting air over the central surface of the cornea to measure the IOP. Britt et al. [10] photographed the noncontact air puff tonometer in their study and observed that the tear film on the ocular surface was broken and crushed by the effect of air pressure at the time of IOP measurement. They also observed many aerosol particles around the eye surface that continued to increase with spray times [10].

On the other hand, Icare tonometers work with the principle of minimal contact with the center of the cornea through the tip of the probe. Icare tonometer (ICT) is a handheld, battery-powered, portable tonometer with rebound technology. During the measurement, the movement parameters and contact time of the probe are analyzed. The tonometer has a light and small plastic tonometer tip and probe, which provides instant contact with the cornea. The touch of the probe is very gentle and short-term; thus, it is hardly noticeable by the patient. When the working principles of the above-mentioned devices are taken into consideration, it is presumable that these devices may have an effect on the tear film.

Tear film stability can be analyzed by non-invasive methods such as tear film lipid layer interferometry, TSAS or grid xeroscopes, or by invasive techniques using fluorescein such as TBUT. In our study, a significant decrease in TBUT was detected at 1 min, 5 min and 10 min compared with baseline values after the measurement with both methods. It was observed that the decrease in mean TBUT values after NCT measurement was significantly higher than the mean decrease observed after ICT. An explanation for this difference is that the ICT affects a smaller area on the corneal surface, whereas NCT affects a relatively larger area. In addition, during NCT, air puff generated from the nozzle reaches the eye from a distance of 11 mm in 2.6 ms of traveling time at an average speed of 5 m/s [11]. This first creates a transient reduction in the air pressure near the eye, causing the accumulated tear film to expand for about 3 ms before the trailing jet from the puff impinges on the cornea [11]. The trailing jet impingement causes corneal deformation and what we believe a “tsunami effect” on the tear film, and possibly a “splash effect” displacing the secretory mucins into the air interface and/or a “crush effect”, altering the structure of membrane bound mucins that are necessary for the wettability of the ocular surface [12,13]. These plausible mechanisms might have induced the tear instability after NCT. ICT induced tear instability at 1, 5 and 10 min to a lesser extent than that observed with the NCT measurement, suggesting that ICT might have caused a local “ripple effect” on the tear film and a limited “crush effect” on membrane bound mucins at the site of corneal contact. Indeed, the ICT plastic probe affects an area with a diameter of 0.8 mm (approximately 0.5024 mm^2^), whereas NCT affects a much larger area with a diameter of 3.6 mm (approximately 11.3354 mm^2^). TSAS measurements of the SRI back up this hypothesis, in that SRI values significantly worsened from baseline to 10 min after NCT whereas no significant changes in SRI were observed with ICT measurements. High-speed filmography of the tear film–corneal interface during NCT and ICT measurements may provide very interesting information in this sense.

Another important observation in this study was that 53% of the eyes attained normal tear stability 10 min after ICT testing whereas only 20% of the eyes exhibited normal tear stability 10 min after NCT testing. These observations have two implications: ① Tear stability measurements such as TBUT or TSAS or other stability examinations should not be performed within 10 min of ICT or NCT. It is reasonable to wait at least 20–30 min for tear stability exams after ICT, and may be slightly longer after NCT examinations; ② It is best to perform tear stability examinations before NCT or ICT in clinical settings. Although these may sound like commonsense suggestions, tear instability in a prospective study of this nature using time-wise observations of SRI or TBUT comparing the effect of two IOP measurement methods have not been performed until now. It would be interesting to compare tear stability changes after Perkins, Goldman or Tonopen measurements in the future. 

One limitation of the current preliminary study is the relatively small number of subjects who consented to the protocol of the study. Our observations should be repeated on a larger number of healthy subjects and subjects with dry eye disease in future investigations. Study protocols enabling a comparison of both eyes with and without IOP measurements to compare the tear stability measurements would add strength to the results of studies of this nature. Inter-eye comparisons could not be performed in the current study because only mono-ocular tests and analyses were allowed according to IRB decisions, which is another limitation. The addition of control groups with no IOP measurements to compare the tear function would provide helpful information as well.

One major limitation of this study was that the long-term effects of IOP measurements on tear stability beyond 10 min were not evaluated due to the IRB decision not to cause a longer wait time in the ophthalmology clinic for subjects who visited for spectacle prescriptions and volunteered to participate in the current study.

## 5. Conclusions

In conclusion, intraocular pressure measurement in routine ophthalmology clinical practices by either NCT or ICT cause the deterioration of tear film stability, which might affect tear stability tests when performed soon after IOP measurements. It is best to wait at least for 20–30 min after the IOP measurement before evaluating the tear film and the corneal surface or the tear film–corneal surface evaluation tests should be performed before IOP measurements in dry eye clinics.

## Figures and Tables

**Figure 1 jcm-11-02819-f001:**
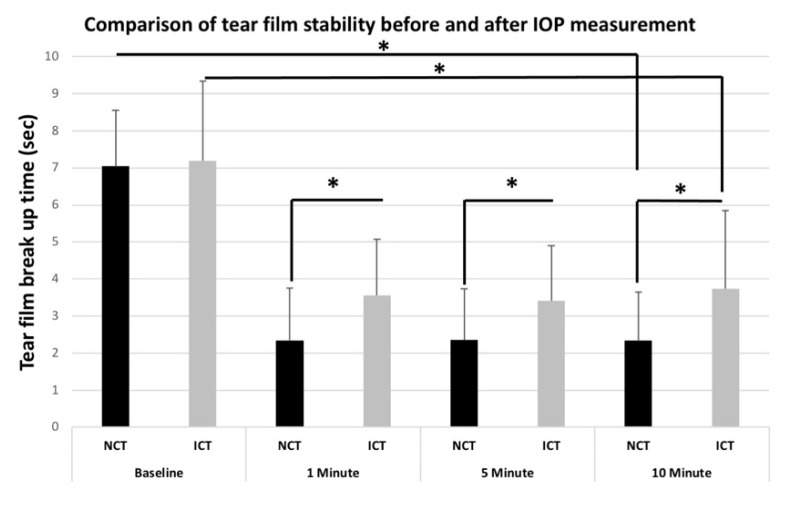
The mean TBUTs showed a significant decrease at 1 min, 5 min and 10 min compared with baseline values for the ICT (*p* < 0.01). NCT-TBUT and ICT-TBUT values are also comparable with each other in the same time period. There were no statistically significant differences between NCT-Baseline and ICT-Baseline TBUT values (*p* > 0.05). Statistically significant differences were found at 1 min, 5 min and 10 min TBUT values between NCT and ICT (*p* < 0.05). * represents *p* < 0.05.

**Figure 2 jcm-11-02819-f002:**
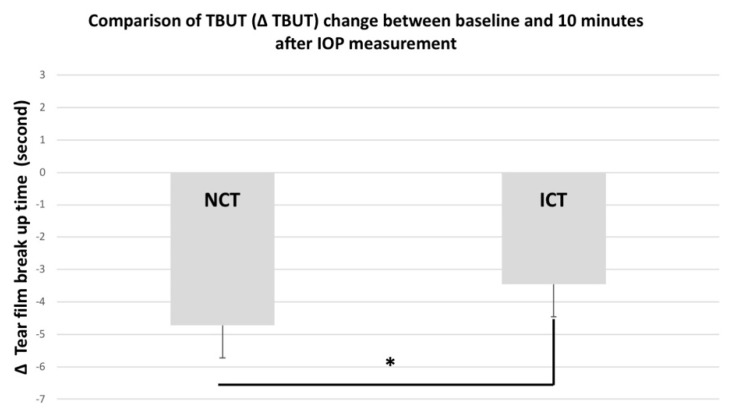
The mean change in TBUT from baseline to 10 min after tonometry (ΔTBUT) was significantly lesser for ICT measurements compared with NCT measurements (*p* < 0.05). * represents *p* < 0.05.

**Table 1 jcm-11-02819-t001:** SRI of subjects before and after IOP measurements. Data are presented as the mean ± SD. SRI, surface regularity index. The time-wise variation in SRI from baseline to 5 and 10 min was observed to be significantly higher for NCT (noncontact tonometry). However, there was no significant difference in terms of SRI for the ICT (Icare tonometry). * *p* < 0.05.

The Mean SRI Values	Baseline	5 min	10 min
**NCT—SRI**	0.96 ± 0.62	1.29 ± 0.72 *	1.25 ± 0.62 *
**ICT—SRI**	1.04 ± 0.49	1.07 ± 0.56	1.03 ± 0.77

## Data Availability

Not applicable.

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
