# Peer review of "The Impact of Noncontact Tonometry and Icare Rebound Tonometry on Tear Stability and Dry Eye Clinical Practice"

_jcm, 2022, doi:10.3390/jcm11102819_

Round 1

Reviewer 1 Report

The manuscript is well written but needs major revision :

1- Number of eyes is too small and should be increased

2- it is a non comparative to the other eye ,comparetive studies with control groups give more power to the study.

3-Results should include more details about the original tear film status in both eyes.

Author Response

Reply to the reviewer’s comments:

We thank the reviewer for his/her favorable view of the MS.

  • We totally agree with the reviewer that 15 eyes is a relatively small number.
  • Please let me clarify the difficulties we encountered with the protocol finalization during the IRB review and during the recruitment process.

Due to nature of the study protocol, the methodology in itself was cumbersome since it required an initial evaluation of tear functions in patients who came for eyeglass prescriptions. Only those subjects with no symptoms and normal tear stability and normal IOPs were invited to join the study. Since tear tests had to be conducted strictly under the same temperature and humidity conditions at the same environment, the IRB imposed the protocol that all testings had to be performed on the same day. After a recruitment process of 8 months, only 10 % of the outpatients(n=15)examined for spectacles agreed to come later on a weekday to undergo the IOP and tear function measurements. Three doodle pools to align these subjects’ schedules were carried out over 2.5 months so that all could come on the same day. The IRB approved study interval and budget was for one year. The recruitment and scheduled alignments took 10.5 months where recruitment had to be stopped and initiate the study and data analyses.

I would have loved to take further efforts to increase the number of subjects but the following circumstances which I hope the reviewers will understand preclude us from doing so:

A) The first author, myself, has resigned from all affiliations in Japan due to acquisition of a Distinguished Talent PR in Ophthalmology in Australia and is in the process of moving to Sydney to assume a new post in Sydney Univ. Dept. Ophthalmology. Dr. Simsek has left Japan to assume an assistant professorship in Turkey. Dr.Takashi Kojima has also resigned from Keio University and went to Nagoya to assume a position in a private clinic. Dr Naohiko Aketa has left Tokyo Dental College and is now working for the Ministry of Health. Kazuo Tsubota has retired from Keio University and is not seeing patients anymore and is functioning as the CEO of a business company he established. Dr. Jun Shimazaki is retiring in March 2023. Therefore with none of the authors left at Tokyo Dental College Dept. of Ophthalmology, it would be impossible for the authors to recruit new subjects.

B) At this point, I have looked into the possibility of asking the new staff members at TDC to conduct additional recruitment. I have also checked with the Ethic Board at TDC on  the current review backlog. An additional recruitment would need another IRB application and they have given me a time frame of 4 months at least to receive a new approval and a possible start in September. The recruitment and conduct of the study process would take at least 6 months, new stat analyses, figure redrawings and rewriting would take another 2 months resulting in an overall delay of 12 months to deliver the review. Yet, I did check with the staff members but they kindly suggested that they are already committed to other projects and would unfortunately not be able to assist us with this request. They have also pointed out that there will be a new professorship election at TDC starting this autumn and the new professor might not endorse this project. Please also note that all study subjects received book gift cheques of 30000JY for their assistance, as determined by the previous IRB. The budget has been used for the current study and there apparently is no further budget from TDC to allocate for the new subjects as I have been told.

C)The current study is preliminary but is carefully conducted and still has value within its own circumstances. I do apologize for not being able to recruit more subjects this time. I however, promise the reviewers to carry out their requests in a new study which will include Goldman and Perkins tonometers in Australia in 2023, with a kick off in the latter half with my already recruited PhDs.

This issue is now emphasized in the revised paper on page 7 lines 232-234: “One limitation of the current preliminary study is the relatively small number of subjects who consented to the protocol of the study. Our observations should be repeated on a larger number of healthy subjects and subjects with dry eye disease in future investigations.”

  • It is a non comparative to the other eye ,comparative studies with control groups give more power to the study.

Reply: I totally agree with this comment. Indeed, the initial protocol submitted to IRB had a control group of 20 subjects who would not undergo IOP measurements which would enable us to compare the tear functions throughout the conduct of the study. We were required to change this since the study would cause a burden of extra testings and an extra commute to the hospital. We then proposed having the other eyes undergoing tear function tests without IOP measurements to have some sort of comparison. This proposal also could not go through the IRB committee since they still thought that the study population coming for eye glasses only need not undergo such testings since the study protocol required the testings be done on the same day at the outpatient clinic where the subjects needed to come early in the morning and wait till late in the afternoon to undergo the tear testings. We were also advised that going outside the hospital with changes in humidity and temperature could have affected the tear functions. We then revised the protocol so that the volunteers came early in the morning before the start of outpatient clinic when we arranged the location of tonometers and TSAS next to each other to ascertain a smooth and quick conduct of testings which were performed at 1, 5 and at 10 minutes. When the morning testings were done, the devices were returned to their original positions and the subjects were requested to move to a lounge on the same floor where we had set the temperature and humidity controls equivalent to those at the outpatient department. The subjects were provided the same boxed lunches and water and were requested to wait for some 6 hours spending a leisure time for their afternoon testings when the routine outpatient clinic was over and the devices were realigned for the purposes of this study. For the sake of avoiding temperature and humidity impact on the tear film, the protocol required a long waiting time in the hospital for the tests. IRB committee decided that the tests that the subjects are to undergo should be as concise and brief as possible which finallyresulted in  a monocular assessment study. Under these restrictions, we could only carry out monocular testings and perform monocular analyses.The need for subjects having to wait in the hospital to avoid temperature and humidity changes is now outlined in the methods. The need for comparison with other eyes increasing the power of such studies is now emphasized in the discussion section on page 7 (L234-240): “Study protocols allowing a comparison of both eyes with and without IOP measurements to compare the tear stability measurements would add strength to the results of studies of this nature.Inter-eye comparisons could not be performed in the current study since only monoocular testings and analyses were allowed according to IRB decision which is another limitation.Addition of control groups with no IOP measurements to compare the tear function would provide helpful information as well.”

Results should include more details about the original tear film status in both eyes.

  • Thank you for this nice comment. The recruitment baseline criteria were stated in the MS as “Subjects underwent OSDI questionnaires, TBUT and fluorescein and Lissamine Green vital stainings for recruitment into the study. Only asymptomatic subjects with OSDI scores <13 points, with vital staining scores <1point, and TBUT scores > 5 seconds were recruited.”

However,since the baseline testings were compared at the first visit on both eyes before recruitment and invitation to study, we were able to provide the details of original tear film data on both eyes. Statistical analyses did not find significant differences between BUT and FS scores between both eyes at baseline. This information is now being provided in the results section. Since the IRB did not allow the conduct of testings on both eyes for the study visit, a comparison between both eyes could not be performed for 1min, 5min and 10 minute tear functions. The relevant information is now provided in the results section on pages 3 and 4 (L120-125): “The mean OSDI score at the time of recruitment into the study was 2.26±2.63 points (minimum:0 point; maximum: 8 points). The mean baseline TBUT at the time of recruitment was 6.92±0.50 seconds for the right eyes and 6.03±0.50 seconds for the left eyes respectively. There were no statistically significant differences between the baseline TBUT values in between both eyes (p>0.05). All eyes had no fluorescein or Lissamine green staining (0 point score) at the baseline measurements which allowed them to be included into the study.

Reviewer 2 Report

The examined case series report is well written, the proposed topic is sufficiently innovative, the scientific analysis is accurate and well executed as well as the statistical evaluation. overall, therefore, there is little to review, however:

1) lissamine green examination for patient's screening is mentioned but results are not reported.
2) an important lack is basal Schirmer test that could have been performed to better evaluate the sample of patients examined.

it would be interesting to increase the sample  of examined patients.

Author Response

Reviewer 2 Comments

The examined case series report is well written, the proposed topic is sufficiently innovative, the scientific analysis is accurate and well executed as well as the statistical evaluation. overall, therefore, there is little to review, however:

1) lissamine green examination for patient's screening is mentioned but results are not reported.
2) an important lack is basal Schirmer test that could have been performed to better evaluate the sample of patients examined.

it would be interesting to increase the sample  of examined patients.

Reply to the reviewer’s comments:

Thank you for your favorable and nice comments in relation to the current work.

1)Indeed, FS and LG testings were done at baseline for recruitment. The recruitment criteria were stated as follows in the MS:” “Subjects underwent OSDI questionnaires, TBUT and fluorescein and Lissamine Green vital stainings for recruitment into the study. Only asymptomatic subjects with OSDI scores <13 points, with vital staining scores <1point, and TBUT scores > 5 seconds were recruited.”

Since both reviewers requested the provision of baseline tear function data these are now given in the results section of the revised MS as follows: ”The mean OSDI score at the time of recruitment into the study was 2.26±2.63 points (minimum:0 point; maximum: 8 points). The mean baseline TBUT at the time of recruitment was 6.92±0.50 seconds for the right eyes and 6.03±0.50 seconds for the left eyes respectively. There were no statistically significant differences between the baseline TBUT values in between both eyes (p>0.05). All eyes had no fluorescein or Lissamine green staining (0 point score) at the baseline measurements which allowed them to be included into the study.”

2)Please note that basal secretion testings would require the instillation of a drop of anesthetic which would dry up the ocular surface and adversely effect the tear stability testings as reported previously in literature related to functional visual acuity. Basal secretion testing influences TSAS measurements in our experience too and were not included in the protocol of the current study.

  • This is an important comment that we agree by heart. The circumstances why we were not able to do this request are clarified below. Please follow: We totally agree with the reviewer that 15 eyes is a relatively small number. Please let me clarify the difficulties we encountered with the protocol finalization during the IRB review and during the recruitment process:

Due to nature of the study protocol, the methodology in itself was cumbersome since it required an initial evaluation of tear functions in patients who came for eyeglass prescriptions. Only those subjects with no symptoms and normal tear stability and normal IOPs were invited to join the study. Since tear tests had to be conducted strictly under the same temperature and humidity conditions at the same environment, the IRB imposed the protocol that all testings had to be performed on the same day. After a recruitment process of 8 months, only 10 % of the outpatients(n=15)examined for spectacles agreed to come later on a weekday to undergo the IOP and tear function measurements. Three doodle pools to align these subjects’ schedules were carried out over 2.5 months so that all could come on the same day. The IRB approved study interval and budget was for one year. The recruitment and scheduled alignments took 10.5 months where recruitment had to be stopped and initiate the study and data analyses.

I would have loved to take further efforts to increase the number of subjects but the following circumstances which I hope the reviewers will understand preclude us from doing so:

A) The first author, myself, has resigned from all affiliations in Japan due to acquisition of a Distinguished Talent PR in Ophthalmology in Australia and is in the process of moving to Sydney to assume a new post in Sydney Univ. Dept. Ophthalmology. Dr. Simsek has left Japan to assume an assistant professorship in Turkey. Dr.Takashi Kojima has also resigned from Keio University and went to Nagoya to assume a position in a private clinic. Dr Naohiko Aketa has left Tokyo Dental College and is now working for the Ministry of Health. Kazuo Tsubota has retired from Keio University and is not seeing patients anymore and is functioning as the CEO of a business company he established. Dr. Jun Shimazaki is retiring in March 2023. Therefore with none of the authors left at Tokyo Dental College Dept. of Ophthalmology, it would be impossible for the authors to recruit new subjects.

B) At this point, I have looked into the possibility of asking the new staff members at

             TDC to conduct additional recruitment. I have also checked with the Ethic Board at      

             TDC on the current review backlog. An additional recruitment would need another 

             IRB application and they have given me a time frame of 4 months at least to receive

             a new approval and a possible start in September. The recruitment and conduct of

             the study process would take at least 6 months, new stat analyses, figure redrawings

             and rewriting would take another 2 months resulting in an overall delay of 12

             months to deliver the review. Yet, I did check with the staff members but they kindly

             suggested that they are already committed to other projects and would

             unfortunately not be able to assist us with this request. They have also pointed out

             that there will be a new professorship election at TDC starting this autumn and the

             new professor might not endorse this project. Please also note that all study subjects

             received book gift cheques of 30000JY for their assistance, as determined by the

             previous IRB. The budget has been used for the current study and there apparently is

             no further budget from TDC to allocate for the new subjects as I have been told.

C)The current study is preliminary but is carefully conducted and still has value within its own circumstances. I do apologize for not being able to recruit more subjects this time. I however, promise the reviewers to carry out their requests in a new study which will include Goldman and Perkins tonometers in Australia in 2023, with a kick off in the latter half.

This issue is now emphasized in the revised paper on page 7 lines 232-234: “One limitation of the current preliminary study is the relatively small number of subjects who consented to the protocol of the study. Our observations should be repeated on a larger number of healthy subjects and subjects with dry eye disease in future investigations.”

Round 2

Reviewer 1 Report

The author improved the quality of the manuscript and accepted for publication in this form